# Dependency-aware action planning for smart home

**Jongjin Kim**[ID], **Jaeri Lee, Jeongin Yun, U. Kang**[ID]*

Data Mining Lab, Seoul National University, Seoul, Korea

* ukang@snu.ac.kr

**Data Availability Statement:** All data are publicly available from the GitHub repository (https://github.com/SmartAid-IoT/SmartAid).

**Funding:** This work is supported by Samsung Electronics co., Ltd. The Institute of Engineering Research at Seoul National University provided

## Abstract

How can a smart home system control a connected device to be in a desired state? Recent developments in the Internet of Things (IoT) technology enable people to control various devices with the smart home system rather than physical contact. Furthermore, smart home systems cooperate with voice assistants such as Bixby or Alexa allowing users to control their devices through voice. In this process, a user's query clarifies the target state of the device rather than the actions to perform. Thus, the smart home system needs to plan a sequence of actions to fulfill the user's needs. However, it is challenging to perform action planning because it needs to handle a large-scale state transition graph of a real-world device, and the complex dependence relationships between capabilities. In this work, we propose SMARTAID (Smart Home Action Planning in awareness of Dependency), an action planning method for smart home systems. To represent the state transition graph, SMARTAID learns models that represent the prerequisite conditions and operations of actions. Then, SMARTAID generates an action plan considering the dependencies between capabilities and actions. Extensive experiments demonstrate that SMARTAID successfully represents a real-world device based on a state transition log and generates an accurate action sequence for a given query.

## Introduction

*How can we manipulate a smart home device to reach a desired state?* The usage of smart home and Internet of Things technologies has increased in recent years, leading to extensive research in utilizing data mining and machine learning methods in the smart home technology field [1–4].

Smart home systems provide a convenient environment to control connected IoT devices for users, increasing their satisfaction and productivity. Furthermore, recent developments in voice assistants such as Bixby or Siri enable smart home systems to provide voice interfaces to control devices [5, 6]. A user clarifies the desired state of the device and the smart home system fulfills the user's need by executing the required actions on the device [7]. Hence, it is important to generate a proper action sequence to meet the goal of the user's query. Currently, either device manufacturers or smart home system providers manually code the solution to find the proper action sequence for each device. However, as the number of IoT devices increases [8],

research facilities for this work. The Institute of Computer Technology at Seoul National University provides research facilities for this study. The funders had no role in the methodology of the study.

**Competing interests:** The authors have declared that no competing interests exist.

it is hard to manually implement the action planning scheme for each device. Thus, it is crucial to construct an algorithm to perform action planning in the smart home system without supervised information, using only the information of target devices and logs from them. This algorithm needs to be fast and efficient enough to be executed by IoT hubs.

The action planning problem for smart home aims to find a sequence of actions that change the state of the device to the target state. Fig 1 shows an example of the action planning problem. In this example, a user wants to change the channel of the TV from '46' to '29' while maintaining the switch to be off. One may try to directly call the 'setChannel' command to fulfill the user's query. However, 'setChannel' command is not callable if the switch of the TV is off as shown in Fig 1-(II). Thus, the naive approach to directly control the target attribute of the device fails to plan the valid action sequence. Meanwhile, Fig 1-(III) considers the dependency of the 'setChannel' command by first turning on the switch, then setting the channel of the TV, and finally turning off the switch again. Like the action sequence planned in Fig 1-(III), we aim to successfully find a valid action sequence that changes the current state of a device to the target state while considering dependencies.

Action planning for smart home encounters three main challenges. First, the number of states of smart home devices is intractable. For instance, consider a simple bulb which has only five capabilities: 'switch', 'switchLevel', 'colorTemperature', 'hue', and 'saturation'. The 'switch' takes a value of either 'on' or 'off'. The rest of the capabilities describe the color of the light and each has about 100 possible values. Hence, even a simple bulb has $2 \times 100^4 = 200$ million possible states and the transition relationships between these states are innumerable. Second, actions are only executable when their prerequisite conditions are satisfied. If a user is currently watching a content from HDMI source on a TV and desires to watch channel 5, the user cannot directly use the command 'setChannel'. This is because channels can only be set when the value of 'mediaInputSource' capability of the TV is set to 'digitalTv'. Third, an action to control a specific value may intrude on other actions to change other values. Assume that a user wants to turn off a bulb and also change the color temperature of the bulb for future use. However, sequentially controlling the 'switch' and 'colorTemperature' capabilities would not

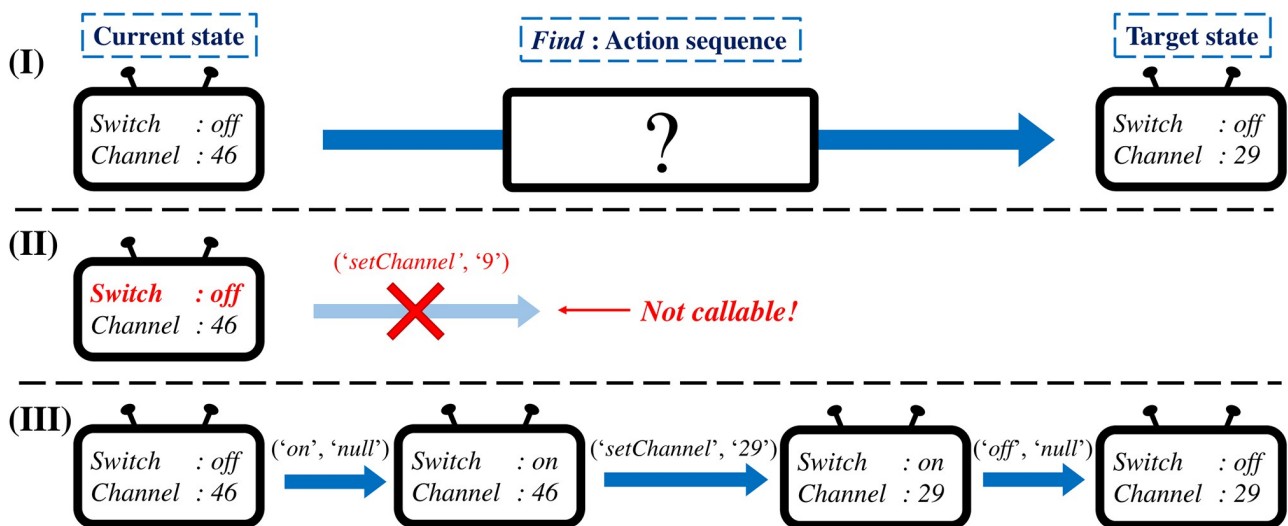

**Fig 1. Action planning problem.** (I) shows an example of the action planning problem on a TV. The user wants to change the channel of the TV from '46' to '29'. In this case, directly calling 'setChannel' command leads to a failure since it is not callable when the switch is off as shown in (II). Our goal is to generate the action sequence shown in (III) by considering the dependency of commands.

accomplish the goal, since we need to turn on the switch again to change the color temperature. Hence, the system must address this intrusion issue to solve the action planning problem.

In this paper, we propose SMARTAID (Smart Home Action Planning in awareness of Dependency), an efficient and accurate approach for action planning for smart home. SMARTAID solves the challenges with the following three main ideas. First, SMARTAID learns simple machine learning models that describe each command of a device without creating a full state transition graph for a device. Second, SMARTAID considers the prerequisite condition of each command and ensures those conditions are met in the action sequences before executing target commands. Third, SMARTAID carefully arranges the setting order of the capabilities not to rechange the controlled value. SMARTAID successfully generates accurate action sequences given any target state with minimal time and memory overhead.

Our contributions are summarized as follows:

- **Problem formulation.** We define the problem of action planning for smart home which has not been addressed before. Action planning for smart home aims to generate a sequence of actions for transitioning to a target state from any given initial state.

- **Method.** We propose SMARTAID, an efficient and accurate method for action planning for smart home. SMARTAID learns simple machine learning models for each command of a device to address the problem of the massive size of the state transition graph. SMARTAID also considers the prerequisite conditions that must be met for each command and ensures they are satisfied before executing the target command to avoid calling it at an uncallable state. SMARTAID filters and orders command so as not to unwillingly change other values that are not the target to change. The codes are available at https://github.com/SmartAid-IoT/SmartAid.

- **Experiment.** Experiments on real-world datasets show that SMARTAID generates an action sequence with up to 85.7% lesser visited states and up to 91.1% lesser states maintained in memory during inference compared to the best competitor. This shows that SMARTAID is the most time-efficient and memory-efficient method to solve the action planning problem for smart home. We also show that SMARTAID accurately learns the command operation of devices.

- **Real-world dataset.** We collect and open-source a dataset for action planning for smart home. The dataset is collected from SmartThings platform, a famous IoT platform with 62 million users. This is the first dataset for studying action planning for smart home. The datasets are available at https://github.com/SmartAid-IoT/SmartAid.

## Action planning for smart home

### Terminology

We define the terminology used in action planning for smart home.

- **Capability.** A capability is a feature of a device such as *'switch', 'volume',* and *'channel'* of a TV. A device has fixed capabilities even if the state of the device changes.

- **Value of capability.** A value of a capability represents the current state of a device. For example, if the value of *'audioVolume'* capability of a TV is *'20'*, meaning that the current volume of the TV is set to *'20'*. Each capability has a pre-defined set of possible values. For instance, the set of possible values of *'audioVolume'* capability is *{'0', '1', . . ., '100'}*, and that of *'mediaInputSource'* capability is *{'digitalTV', 'HDMI'}*.

**Table 1. Capability statistics of TV.** A device is specified with a set of capabilities, values, and commands.

| | Capability | Value of capability | Command |
|---|---|---|---|
| | switch | on, off | on, off |
| | audioVolume | 0-100 | volumeUp, volumeDown, setVolume |
| | audioMute | muted, unmuted | mute, unmute, setMute |
| | tvChannel | 506—839 | channelUp, channelDown, setChannel |
| | mediaInputSource | digitalTv, HDMI | setInputSource |
| **Total** | **5** | **140** | **12** |

- **State.** A state of a device is a dictionary that defines the current condition of the device with capabilities and their values as key-value pairs. For instance, if the state of a TV is *{'switch':'on', 'tvChannel':'42', 'audioVolume':'20', 'mediaInputSource':'digitalTV'}*, the TV is currently on channel *'42'* with volume *'20'*.

- **Command.** A command is a function that takes an argument and a current state as an input and returns the next state as an output. For a command $m$, $m(a, s_1) = s_2$ means that if we execute the command $m$ on the device in the state $s_1$ with an argument $a$, the device would change into state $s_2$. Each command takes a different form of argument, and a command that does not require a specific argument has *'null'* as an argument. For example, *'setVolume'* command takes an integer between 0 to 100 as an argument while *'on'* takes only *'null'* as an argument.

- **Action.** An action is a pair of a command and an argument. It represents a single step in a consecutive control over a device. We denote an action of a command $m$ with an argument $a$ as $(m, a)$. For instance, *('setVolume', '20')* is an action that changes the volume of a TV to *'20'* using the *'setVolume'* command and *('on', 'null')* is an action that turns on a device.

A device is specified by its capabilities, possible values of each capability, and commands. A system identifies the state of a device through its current values for capabilities. Then, it controls the device by sending commands and proper arguments to execute. We borrow terminologies from SmartThings but the general framework is identical for other smart home systems such as Google Alexa or Apple Home.

We give an example of the terms in Table 1 which shows the specifications of a TV. The TV has five capabilities, 140 possible values, and 12 commands to control. The total number of the states of TV is $2 \times 101 \times 2 \times 33 \times 2 = 26664$, which is the product of the number of possible values of each capability. Note that there are significantly more states compared to the number of capabilities and their possible values.

### Action planning

The goal of the action planning for smart home is to generate an action sequence that changes the current state of the device to the target state. We formally define the problem as follows.

**Problem 1** (Action Planning for Smart Home). ***Given*** *the set* $\mathbf{P} = \{P_1, P_2, \ldots, P_{n_p}\}$ *of capabilities, the sets* $\mathbf{V}_1, \ldots, \mathbf{V}_{n_p}$ *of possible values for each capability, the set* $\mathbf{M} = \{M_1, M_2, \ldots, M_{n_M}\}$ *of commands, the current state* $S_c$, *the target state* $S_t$, *and the set* $\mathbf{L}$ *of state transition logs whose element* $(S^{prev}, m, a, S^{next}) \in \mathbf{L}$ *indicates* $S^{next} = m(a, S^{prev})$ *for an action* $(m, a)$, ***Find*** *an action sequence* $[(m_1, a_1), \ldots, (m_k, a_k)]$ *such that* $m_k(a_k, (\ldots, m_1(a_1, S_c))) = S_t$ *where* $k$ *is the length of the sequence,* $m_1, \cdots, m_k$ *are commands, and* $a_1, \cdots, a_k$ *are arguments.*

For instance, consider the action planning problem illustrated in Fig 1 for the TV specified in Table 1. There are 5 capabilities and 12 commands in the TV, thus $n_P$ = 5 and $n_M$ = 12. The set **P** of capabilities is *{'switch', 'audioVolume', 'audioMute', 'tvChannel', 'mediaInputSource'}* and the set **M** of commands is *{'on', 'off', 'volumeUp', 'volumeDown', 'setVolume', 'setMute', 'mute', 'unmute', 'channelUp', 'channelDown', 'setChannel', 'setInputSource'}*. The sets $V_1$, ..., $V_5$ of possible values for each capability are *{'on', 'off'}*, *{'0', '1', ···, '100'}*, *{'muted', 'unmuted'}*, the set of available tv channels, and *{'digitalTV', 'HDMI'}* as shown in the 'Value of capability' column in Table 1. The current state $S_c$ and the target state $S_t$ in Fig 1-(I) are *{'switch':'off', 'tvChannel':'46'}* and *{'switch':'off', 'tvChannel':'29'}*, respectively, where we omit unnecessary capabilities for brevity. Thus, we need to find an action sequence that changes the value of *'tvChannel'* to *'29'* to solve the action planning problem. The answer shown in Fig 1-(III) is *[('on', 'null'), ('setChannel', '29'), ('off', 'null')]*.

## State transition graph

The action planning problem is equivalent to the path-finding problem on the state transition graph of the target device. The state transition graph of a device is a directed graph that specifies the functionality of the device. Each node of the state transition graph represents a possible state of the device, and each edge of the graph represents an action that changes the state of the device from its starting point to the endpoint. Note that multiple edges can connect the same pair of nodes. For instance, both (*'setVolume', '24'*) and (*'volumeUp', 'null'*) actions modify the TV's volume from *'23'* to *'24'*, so that the two nodes representing volume *'23'* and volume *'24'* are connected by both the (*'setVolume', '24'*) edge and the (*'volumeUp', 'null'*) edge.

Fig 2 shows a part of the state transition graph for a TV with only two capabilities: *'tvChannel'* and *'audioVolume'*. Note that the state transition graph is massive even with a small number of capabilities since the number of possible states increases exponentially as the number of

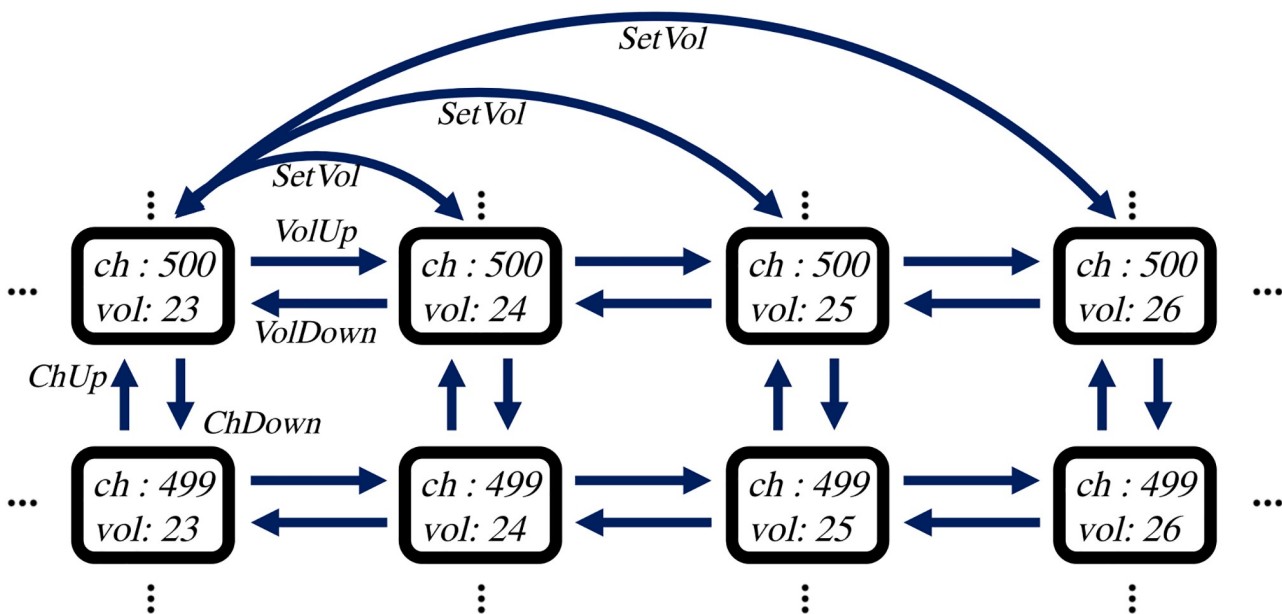

**Fig 2. State transition graph of a TV with only two capabilities.** *'ch'* and *'vol'* represent *'tvChannel'* and *'audioVolume'* capabilities, respectively. *'SetVol', 'VolUp', 'VolDown', 'ChUp',* and *'ChDown'* represent *'setVolume', 'volumeUp', 'volumeDown', 'channelUp',* and *'channelDown'* commands, respectively. The size of the state transition graph is massive even though the number of capabilities is small.

capabilities grows. Hence, the state transition graphs of real-world devices are intractable and cannot be stored.

Thus, it is important to capture necessary information of the state transition graph from the log data to solve the action planning problem.

## Proposed method

In this section, we propose SMARTAID (<u>Smart</u> Home <u>A</u>ction Planning <u>i</u>n awareness of <u>D</u>ependency), an efficient and accurate method to solve the action planning problem.

### Overview

We address the following challenges to perform action planning for smart home:

- **Intractable transition graph.** There is no explicit state transition graph for a given device due to its massive scale. How can we represent the state transition graph of the given device?

- **Complex dependency of commands.** Commands to control the value of the target capability are not always callable depending on the values of other capabilities. How can we properly set the value of the target capability regardless of the given state?

- **Interrupting commands.** Latter commands of an action sequence might change already set values of capabilities. How can we avoid intrusion while planning an action sequence?

The main ideas of SMARTAID are summarized as follows:

- **Compact representation focusing on commands.** We represent the state transition graph by learning simple machine learning models that describe the commands of the device. This allows us to perform action planning without fully constructing or storing the graph.

- **Condition matching before execution.** We find an action sequence to make the desired command executable. Thus, we construct a valid action plan even if the necessary command is not callable in the current state.

- **Carefully ordered capabilities.** We carefully arrange the setting-order of the capabilities to prevent latter actions from overriding previously executed actions.

Fig 3 shows the overall process of SMARTAID. SMARTAID consists of two phases: graph construction phase and action planning phase. In the graph construction phase, the state transition log data of an IoT device is given to SMARTAID. Then, SMARTAID summarizes the state

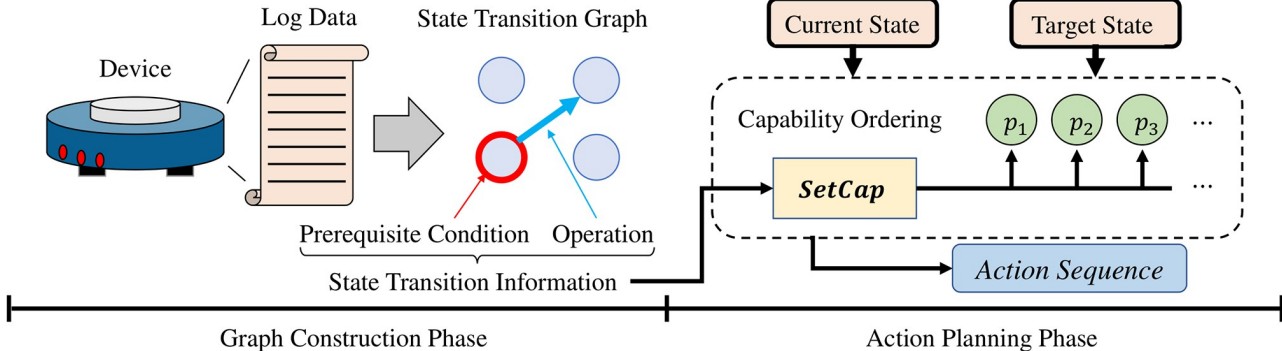

**Fig 3. Overall process of SMARTAID.**

transition graph of the device by learning the prerequisite conditions and operations of actions from the given data. In the action planning phase, a current state and a target state of the device are given. Finally, SMARTAID finds a sequence of actions to change the state of the device from the current state to the target state based on the state transition information learned in the graph construction phase.

## Graph construction phase

The goal of the graph construction phase is to learn the necessary graph information to reconstruct the state transition graph of a device from the given state transition log of the device. To reconstruct the state transition graph, we need to know where an edge starts from and where an edge points to. Note that each edge represents an action performed on the device in the state transition graph. Hence, the starting point of an edge is the state in which the command can be executed, and the node pointed by an edge is the state after the action is performed. Thus, SMARTAID learns 1) prerequisite conditions of each command to determine from where the edges corresponding to the command start, and 2) the operation of each command to determine where the edges corresponding to the command point to. For example, consider *'setVolume'* command of a TV. This command is executable if and only if *'switch'* capability of the TV is set to *'on'*, and changes *'volume'* capability of the TV to the specified argument after execution. SMARTAID summarizes *'setVolume'* command as follows:1) the prerequisite condition of *'setVolume'* is when *'switch'* capability is set to *'on'*, and 2) the operation of *'setVolume'* is to change the value of *'volume'* capability to the argument of the action. In this way, all edges corresponding to *'setVolume'* are reconstructable by only these two pieces of information. Consequently, the graph construction phase of SMARTAID returns the state transition information $\mathcal{G}$, a collection of prerequisite conditions and operations of all commands.

**Prerequisite condition learning.** How can we learn the prerequisite condition of a command from the state transition log of the device? We observe that the state transition log contains only successfully executed actions, so the previous states in the log are guaranteed to meet the prerequisite condition of the command. Furthermore, all possible previous states appear on the log if the number of instances in the log is sufficient.

Hence, SMARTAID collects all values of capabilities that enable each command to be executed to extract the prerequisite condition of the command. In the example shown in Fig 4, after SMARTAID gathers all instances of the execution of *'setVolume'* command from the log data, SMARTAID investigates the previous values of each capability to find the prerequisite condition of *'setVolume'*. We observe that only *'on'* appears in the previous value of *'switch'* capability, while all possible values appear in the previous values of *'volume'* capability. Thus, the prerequisite condition to call *'setVolume'* is to set *'switch'* to *'on'*.

SMARTAID utilizes a dictionary to represent the prerequisite condition of an action. A prerequisite condition $C$ of an action $(m, a)$ is a dictionary whose keys are capabilities that are the prerequisite condition to execute $(m, a)$, and $C[p]$ is the list of possible values of the capability $p$ to execute $(m, a)$. For instance, the prerequisite condition of the action (*'setVolume'*, *'20'*) is {*'switch'*:[*'on'*]}

**Operation learning.** How can we learn the operation of a command from the state transition log of a device? A value of a capability after an action is determined by 1) the state before the action and 2) the argument of the action. We train a machine learning model to learn the relationship between the previous state and the next state.

To learn the operation of a target command, SMARTAID collects all instances of the target command from the log data. Then, SMARTAID treats 1) the previous state and the argument as independent variables, and 2) the next value of the capability as a dependent variable to train a

| Prev. switch | Prev. volume | Argument | Next switch | Next volume |
|:---:|:---:|:---:|:---:|:---:|
| on | 20 | 10 | on | 10 |
| on | 10 | 20 | on | 20 |
| on | 50 | 25 | on | 25 |
| on | 45 | 60 | on | 60 |
| … | … | … | … | … |

'*setVolume*' is callable only if the switch is set to '*on*'

**Fig 4. Prerequisite condition learning process for '*setVolume*' command of a TV.** '*setVolume*' command is callable if and only if '*switch*' capability is set to '*on*'.

machine learning model. SMARTAID repeats this process for each capability to learn the full operation of the target command.

Fig 5 illustrates the training process of SMARTAID to learn the operation of '*setVolume*' command for '*volume*' capability of a TV. The previous value of '*volume*' and the argument of the command are treated as independent variables, while the next value of '*volume*' is considered as a dependent variable. Since '*volume*' capability takes a number as its value, SMARTAID applies a regression model to learn the function of '*setVolume*' command. If we use a linear regression,

| | **x** | | | *y* |
|:---:|:---:|:---:|:---:|:---:|
| Prev. switch | Prev. volume | Argument | Next switch | Next volume |
| on | 20 | 10 | on | 10 |
| on | 10 | 20 | on | 20 |
| on | 50 | 25 | on | 25 |
| on | 45 | 60 | on | 60 |
| … | … | … | … | … |

➜ '*setVolume*' : $y = \begin{bmatrix} 0 & 1 \end{bmatrix} \mathbf{x}$

**Fig 5. Operation learning process for '*setVolume*' command of a TV.** '*volume*' of the TV after an action with '*setVolume*' command is determined by the argument of the command.

the model to describe the function of *'setVolume'* command is $y = [0 \ 1]\mathbf{x}$ where $y$ is the next value of *'volume'* capability and $\mathbf{x}$ is a column vector composed of the previous value of *'volume'* and the argument of the command.

## Action planning phase

The goal of the action planning phase is to find an action sequence to change the state of a device to a target state when the current state and the target state are given. A naive approach to solve this problem is to set each capability to the value in the target state one by one using setter commands such as *'setVolume'* or *'setChannel'*. However, there are two challenges to this approach. First, even if we set a capability to the target state, it may change while setting the next capability. For example, assume that we set *'switch'* capability to *'off'*. If we change *'volume'* capability afterward, we need to turn on the switch again to execute volume-related commands such as *'setVolume'*, *'volumeUp'*, or *'volumeDown'*. Second, setter commands are not always available because of complex dependencies between capabilities. For instance, *'setVolume'* command is not callable if *'switch'* capability is set to *'off'*. Hence, we cannot simply call *'setVolume'* command to change *'volume'* if *'switch'* is set to *'off'* as shown in Fig 1. To address these challenges, we first design an algorithm to find an action sequence to reach the target state considering the setting-order of capabilities. Then, we propose a capability-setting function that sets a value of a target capability to a target value considering the prerequisite conditions.

**State-setting.** The goal of the state-setting process is to find an action sequence to set all capabilities to values in the target state given the current state and the target state of a device. We use a function *SetCap* to find an action sequence to set a target capability to a target value considering dependencies between capabilities. *SetCap* takes a current state, a target capability, a target value, the dependency order of capabilities, and the state transition information learned from the graph construction phase as arguments. Then, it returns an action sequence to set the target capability to a target value and the state after the action sequence. Note that values of capabilities that are prerequisite conditions to change the target capability also change as well. See the next section for detailed implementation of *SetCap*.

Given the *SetCap* function, a naive approach for state-setting is repeatedly calling *SetCap* for each capability until the current state equals to the target state. However, this approach leads to an inefficiency since it might set the same capability multiple times; an action sequence found by *SetCap* changes not only the value of the target capability $p_t$ but also the values of prerequisite conditions. For instance, assume that we want to set the TV's *'switch'* to *'off'* and its *'volume'* to *'20'*. The current state of the TV is {*'switch'*:*'on'*, *'volume'*:*'30'*}. If we set *'switch'* capability to *'off'* before setting *'volume'*, we need to turn on the switch again to execute *'setVolume'* command to control *'volume'* capability. Then, we have to set *'switch'* capability to *'off'* again afterward, which is a severe inefficiency. How can we avoid such inefficiency?

SMARTAID sorts capabilities based on dependency relationships to find the order of capabilities to prevent repeatedly setting the same capability. Assume that capabilities are sorted so that prerequisite conditions of an action to change a capability $p$ always appear before $p$ in the order. Then, *SetCap* targeting the capability $p_t$ changes only values of capabilities that appear before $p_t$ in the order, since *SetCap* changes only the target capability and prerequisite conditions. Hence, if we set capabilities by calling *SetCap* in the reverse order, this ensures that the formerly set capabilities are not modified by the subsequent action sequences to set other capabilities.

Thus, finding the order of capabilities according to their dependency is essential for state-setting. SMARTAID utilizes topological sorting to find such ordering. SMARTAID generates a

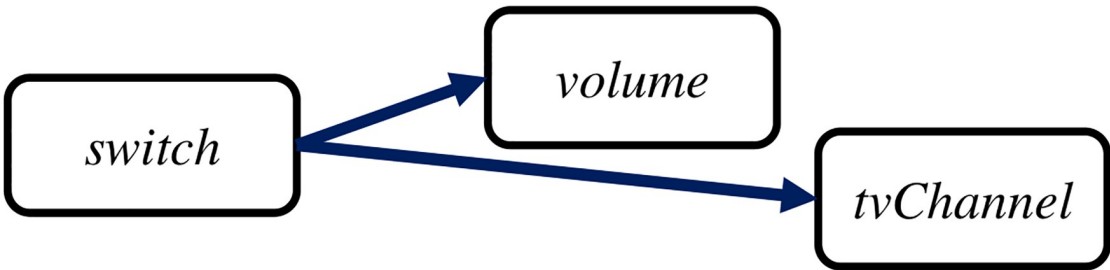

**Fig 6. Dependency graph for a TV.** *'switch'* capability of the TV is a prerequisite condition to control *'volume'* and *'channel'* of the TV.

dependency graph $G_d$ which is a directed graph with capabilities as vertices and prerequisite condition relationships as edges. A parent node in this graph is a prerequisite condition of an action to change its child nodes. Then, SMARTAID orders capabilities in their topological order. Fig 6 shows an example of capability-ordering for a TV. *'switch'* node is the parent of *'volume'* and *'tvChannel'* nodes since *'switch'* of the TV should be *'on'* to control those two capabilities with commands such as *'setVolume'* or *'setChannel'*.

SMARTAID also merges capabilities that form a cycle into a single capability to guarantee that there exists a topological order of capabilities. For instance, a bulb has *'hue'* and *'saturation'* capabilities which are both for controlling the color of light. Whenever a user changes the color by adjusting one of those capabilities, the other capability is automatically set to zero. Hence, they are both dependent on each other and form a cycle in the dependency graph. In this case, we merge the two capabilities into a new capability *('hue', 'saturation')*. Hence, all prerequisite conditions of an action to change a capability $p$ always appear before $p$ since they are the parents of $p$ in the dependency graph.

Algorithm 1 shows the process of state-setting. From lines 1 to 2, SMARTAID determines the order of capabilities to set. In case of the TV shown in Fig 6, the ordering $\mathbf{O}_d$ is [*'switch'*, *'volume'*, *'tvChannel'*]. From lines 4 to 7, SMARTAID sets capabilities one by one in the reverse order of $\mathbf{O}_d$. In the case of the TV, SMARTAID sets capabilities in the following order: *'tvChannel'*, *'volume'*, and *'switch'*. In this way, SMARTAID does not need to repeatedly control formerly set capabilities, thereby avoiding inefficiency. For each capability, SMARTAID finds an action sequence to set the capability with the function *SetCap*. Then, SMARTAID applies the found action sequence to the current state and adds the sequence to the action sequence $\mathbf{A}$. In line 8, SMARTAID returns the found action sequence $\mathbf{A}$ to change the current state $S_c$ to the target state $S_t$. We analyze the complexity of Algorithm 1 in the next section.

**Algorithm 1** Action planning

```
Input: Current state S_c, Target state S_t, the set P of all capabili-
       ties, and the state transition information G learned from the
       graph construction phase
Output: Action sequence A to change S_c to S_t
  1: Draw the dependency graph G_d of all capabilities P.
  2: Let O_d = [p_1, p_2, ..., p_{n_p}] be the topological order of capabilities in
     G_d.
  3: Initialize A as an empty list.
  4: for i in [n_p, ..., 1] do
  5:    R, S_c ← SetCap(S_c, p_t, v_t, O_d, G)
  6:    A.append(R)
  7: end for
  8: return A
```

**Capability-setting.** The goal of the capability-setting algorithm *SetCap* is to change the value of the target capability to the value in the target state when the current state, the target capability, the target value, and the state transition information are given. Our idea to solve this problem is first, to find an action sequence to set the capability ignoring prerequisite conditions, and then insert subsequences to meet the prerequisite condition of each action before execution. For instance, assume that we want to set *'channel'* capability of a TV from *'46'* to *'29'* as shown in Fig 1. SMARTAID first finds an action sequence to set *'channel'* capability ignoring prerequisite conditions, which is [*('setChannel', '29')*]. Then, SMARTAID inserts a sequence of actions to meet the prerequisite condition of *'setChannel'* command before the action.

Algorithm 2 shows the capability-setting algorithm *SetCap* of SMARTAID. In the case shown in Fig 1, the current state $S_c$ is *{'switch':'off', 'tvChannel':'46'}*, the target capability $p_t$ is *'tvChannel'*, and the target value $v_t$ is '29'. The output of the algorithm is the action sequence to change *'tvChannel'* capability to *'29'*, and the state after changing *'tvChannel'* capability. In line 1, *SetCap* looks up the operations of commands from the state transition information $\mathcal{G}$ learned in the graph construction phase, and collects the commands $\mathbf{M}^{(p_t)}$ that change the value of the target capability. In the example, $\mathbf{M}^{(p_t)}$ is *{'setChannel', 'channelUp', 'channelDown'}* since they are only commands whose operations change the target capability *'tvChannel'*. From lines 2 to 3, the algorithm performs BFS (Breadth-First Search) on the value transition graph $G_v^{(p_t)}$ whose vertices are the possible values of $p_t$ and edges are actions of commands in $\mathbf{M}^{(p_t)}$ as shown in Fig 7. Then, the algorithm obtains the shortest action sequence $\mathbf{A}$ = [*('setChannel', '29')*] to change the value of $p_t$ to $v_t$ without considering capabilities other than $p_t$ in line 4. From lines 5 to 17, the algorithm adjusts the action sequence $\mathbf{A}$ to meet the prerequisite condition of each action. For each action $(m_i, a_i)$ in $\mathbf{A}$, *SetCap* first collects the prerequisite conditions to run the action $(m_i, a_i)$ from the state transition information $\mathcal{G}$. Then, for each capability $p$, it finds the action sequence $\mathbf{R}$ to change the value of $p$ to meet the prerequisite condition of $(m_i, a_i)$ by recursively calling *SetCap*. Note that in line 7, *SetCap* orders prerequisite conditions $\mathbf{C}$ to the reverse order of $\mathbf{O}_d$. This ensures that condition matching for latter capability does not interrupt the already matched conditions. In the example, the algorithm calls $SetCap(S_c, 'switch', 'on', \mathbf{O}_d, \mathcal{G})$ and obtains $\mathbf{R}$ = [*('on', 'null')*] since *'switch':'on'* is the prerequisite condition of (*'setChannel', '29'*). Note that the command *'on'* does not require a specific argument, so the argument is *'null'*. Then, the algorithm inserts $\mathbf{R}$ before the action $(m_i, a_i)$ in $\mathbf{A}$ and updates the current state $S_c$ by applying each action in $\mathbf{R}$. In the example, the algorithm updates $S_c$ to *{'switch':'on', 'tvChannel':'46'}*, and update $\mathbf{A}$ to [*('on', 'null'), ('setChannel', '29')*]. The algorithm repeats this process until all capabilities satisfy the prerequisite condition of $(m_i, a_i)$, and then applies $(m_i, a_i)$ to $S_c$. In the example, the algorithm would update $S_c$ to *{'switch':'on',*

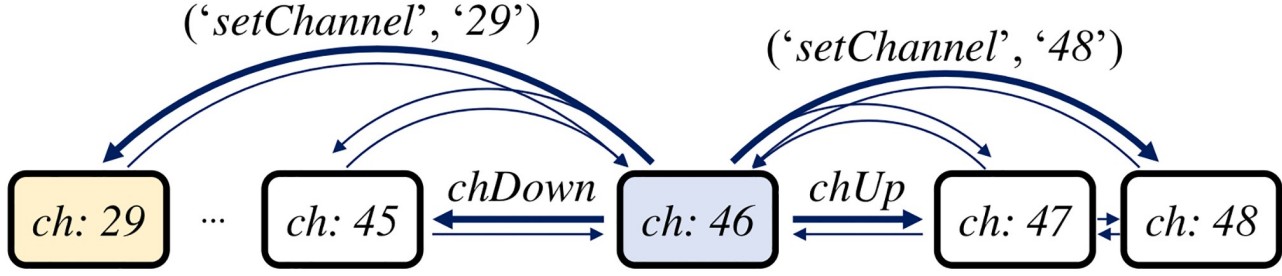

**Fig 7. Value transition graph of a TV for capability 'tvChannel'.** *'ch'*, *'chUp'* and *'chDown'* stand for *'tvChannel'*, *'channelUp'*, and *'channelDown'*, respectively.

*'tvChannel':'29'}* since there is no more prerequisite condition for (*'setChannel', '29'*). In line 18, the algorithm returns the action sequence **A** = [(*'on', 'null'*), (*'setChannel', '29'*)] to change the target capability $p_t$ to the target value $v_t$, and the state after applying the action sequence.

**Algorithm 2** Capability-setting via prerequisite condition matching

```
Input:  Current state S_c, target capability p_t, target value v_t, the
        dependency order O_d of capabilities, and the state transition
        information G learned from the graph construction phase
Output: Action sequence A to change the value of p_t to v_t, and the
        state after applying A to S_c
 1: Function SetCap(S_c, p_t, v_t, O_d, G):
 2:   Find the set M^(p_t) of commands changing p_t in G
 3:   Generate a value transition graph G_v^(p_t) whose vertices are possible
        values of p_t and edges are actions of M^(p_t)
 4:   Find the shortest action sequence A = [(m_1, a_1), (m_2, a_2), ···, (m_k,
        a_k)] of length k from the current value of p_t in S_c to v_t on G_v^(p_t)
        using BFS
 5:   for i in [1, 2, ···, k] do
 6:     C← the prerequisite conditions of (m_i, a_i) in G
 7:     Sort C to the reverse order of O_d
 8:     for each capability p in C do
 9:       if p ∉ C or S_c[p]∈C[p] then
10:          continue
11:       end if
12:       Randomly select v from C[p]
13:       R, S_c ← SetCap(S_c, p, v, O_d, G)
14:       Insert R before (m_i, a_i) in A
15:     end for
16:     S_c ← m_i(a_i, S_c)
17:   end for
18:   return A, S_c
```

## Complexity analysis

We analyze the time and space complexities of the action planning in SMARTAID (Algorithm 2).

**Lemma 1**. *Let $n_P$ be the number of all capabilities. Let $r$ be the maximum length of the path in the dependency graph $G_d$ (see* Fig 6*). Then, the number of calls of the function SetCap during the action planning is at most $O(n_p{}^{r+1})$.*

*Proof*. Assume that function *SetCap* targeting a capability $p$ is called. Then, *SetCap* is recursively called for each capability connected to $p$ in the dependency graph $G_d$. Hence, the total number of recursive calls of *SetCap* when *SetCap* targeting $p$ is called equals the number of paths to $p$ in $G_d$. There are up to $n_p^r + \ldots + n_p = O(n_p^r)$ paths since the maximum length of the path is $r$. The action planning phase of SMARTAID calls *SetCap* for each capability to plan actions as shown in Algorithm 2. Thus, the total number of calls of *SetCap* is $O(n_p^r) \times n_p = O(n_p^{r+1})$.

Note that the length of the path in the dependency is the depth of dependency between capabilities. The maximum depth of dependency is 2 in real-world devices used in our experiments.

**Theorem 1**. *Let $n_P$ and $n_M$ be the numbers of all capabilities and commands, respectively. Let $r$ be the maximum length of the path in the dependency graph. Let $\mathbf{V}_1, \ldots \mathbf{V}_{n_P}$ be the sets of all possible values for each capability, respectively. The time complexity of* SMARTAID *is $O(\max_i |\mathbf{V}_i|^2 \times n_M n_p^{r+1})$*

*Proof*. The bottleneck of function *SetCap* is the BFS on the value transition graph $G_v^{(p_t)}$. The number of vertices in $G_v^{(p_t)}$ is $O(\max_i |\mathbf{V}_i|)$ since each vertex represents a possible value of the

target capability. The number of edges in $G_v^{(p_t)}$ is $O(\max_i|\mathbf{V}_i|^2 \times n_M)$ since each pair of nodes can be connected by up to $n_M$ different commands. Hence, the BFS process has $O(\max_i|\mathbf{V}_i|^2 n_M)$ time complexity. Meanwhile, the number of calls of *SetCap* during the action planning is $O(n_P^{r+1})$ by Lemma 1. Thus, the time complexity of SMARTAID is $O(\max_i|\mathbf{V}_i|^2 \times n_M n_P^{r+1})$.

**Theorem 2** *Let $n_P$ be the number of all capabilities. Let r be the maximum length of the path in the dependency graph $G_d$. Let $\mathbf{V}_1, \ldots \mathbf{V}_{n_p}$ be the sets of all possible values for each capability, respectively. The space complexity of* SMARTAID *is* $O(\max_i|\mathbf{V}_i| \times n_P^{r+1})$

*Proof. SetCap* requires memory for 1) the BFS on the value transition graph $G_v^{(p_t)}$ in Algorithm 2 and 2) storing planned action sequence $\mathbf{A}$. The memory usage during a BFS is $O(\max_i|\mathbf{V}_i|)$ since the number of vertices in $G_v^{(p_t)}$ is the number of possible values of the target capability. The length of the action sequence found by BFS is also $O(\max_i|\mathbf{V}_i|)$ since the maximum length of the path is the number of vertices. Note that the number of calls of *SetCap* during the action planning is $O(n_P^{r+1})$ by Lemma 1. Thus, the space complexity of SMARTAID is $O(\max_i|\mathbf{V}_i| \times n_P^{r+1})$

## Experiments

In this section, we perform experiments to answer the following questions:

- Q1. **Action planning.** How effectively does SMARTAID plan actions for the given inputs?

- Q2. **Operation learning models.** Which machine learning model is the best option for the graph construction phase of SMARTAID?

- Q3. **Case study.** How does SMARTAID work in special cases compared to other methods?

### Experimental settings

**Dataset.** We collect state transition log data using Samsung SmartThings API. We use a bulb (Purecoach LB806 HomeIoT RGB LED smart bulb), a TV (Samsung 2022 QLED 4K TV), a robot cleaner (Samsung BESPOKE Jet Bot), and an air purifier (Samsun BESPOKE Cube) as target devices, as shown in Table 2. For each device, we repeat sending a randomly selected action. Then, we record the previous state, the executed action, and the next state of the device. We record only the controllable capabilities of each device. Table 3 shows an example of the state transition log data. The previous state column and the next state column show the value of each capability before and after executing an action, respectively. The action column shows the command and its argument of the executed action. Note that the arguments of the first and the last instance are null since *'on'* command and *'off'* command do not take arguments.

**Experimental process.** All the methods are implemented in Python and executed on an Intel Core i9-11900F processor. We randomly split each dataset into a training set and a test

**Table 2. Statistics of collected datasets.**

| Device | #Capability | #Value | #Command | #Instance |
|---|---|---|---|---|
| Bulb | 5 | 448 | 6 | 13655 |
| TV | 5 | 140 | 12 | 12517 |
| Robot cleaner | 5 | 16 | 13 | 2154 |
| Air purifier | 4 | 2416 | 12 | 15118 |

**Table 3. Examples of state transition log data collected from a bulb.** 'Level', 'Temp.', and 'Sat.' denote the 'SwitchLevel', 'colorTemperature', and 'Saturation' capabilities, respectively.

| Previous state | | | | | Action | | Next state | | | | |
|---|---|---|---|---|---|---|---|---|---|---|---|
| Switch | Level | Temp. | Hue | Sat. | Command | Arguments | Switch | Level | Temp. | Hue | Sat. |
| off | 40 | 2000 | 0 | 91 | on | null | on | 40 | 2000 | 0 | 91 |
| on | 40 | 2000 | 0 | 91 | setLevel | 70 | on | 70 | 2000 | 0 | 91 |
| on | 70 | 2000 | 0 | 91 | setHue | 20 | on | 70 | 2000 | 20 | 0 |
| on | 70 | 2000 | 20 | 0 | setColorTemperature | 2300 | on | 70 | 2300 | 20 | 0 |
| on | 70 | 2300 | 20 | 0 | off | null | off | 70 | 2300 | 20 | 0 |

set in an 8:2 ratio for training machine learning models in SMARTAID. All machine learning models are implemented with Pytorch and scikit-learn libraries. We use Adam optimizer [9] with a learning rate of 0.01 and L1 regularization coefficient of 0.7 to optimize machine learning models.

## Action planning

To examine the efficiency of action planning, we compare SMARTAID and other path-finding algorithms: BFS, A*, and improved A* [10]. BFS and A* are commonly used general path-finding algorithms. We use Hamming distance which counts the number of different capabilities between two states for a heuristic function of the A* algorithm. Improved A* [10] is a recent improvement of the traditional A* algorithm, which considers the direction of search to perform efficient searching. We randomly generate current states and execute randomly selected actions to generate target states. We change the distance between the current state and the target state from 1 to 5. We generate 1000 inputs for each device and distance.

We report the average number of visited states and the average of the maximum number of states maintained in memory during inference for each experiment to compare the efficiency of the methods. The number of visited states is proportional to the time complexity of the algorithmand the number of memorizing states is proportional to the space complexity of the algorithm. Thus, a smaller number of visited states and a smaller number of states maintained in memory indicate a more efficient method.

Table 4 shows the results. SMARTAID solves the action planning problem with the minimum number of visited states in all cases. This is because the search space of each capability-setting process in SMARTAID is smaller compared to other methodssince SMARTAID sequentially sets each capability one by one as shown in Algorithm 2. Meanwhile, other baselines search the target state from all possible states of the device, which is inefficient. Note that improved A* performs worse than the traditional A* algorithm. This is because the capability of a device is often set by a single command so the search direction is not essential in the action planning problem for smart home. Thus, the heuristic of improved A* consumes more resources without contributing to performance. Hence, SMARTAID is the most time-efficient method since the number of visited states is proportional to the running time of a method.

SMARTAID also requires the smallest memory to store the states, compared to other baselines. This is because SMARTAID does not need to maintain the previously visited states to set up a capability after setting it since the search spaces to set up different capabilities are different in SMARTAID. However, BFS, A*, and improved A* have to store all of the previously visited states which leads to more memory consumption during inference. Thus, SMARTAID is the most space-efficient method since the number of states maintained in memory during inference is proportional to the memory consumption of a method.

**Table 4. Efficiency comparison of action planning methods.** 'Visited' columns show the number of visited states during inference. 'Memory' columns show the maximum number of states maintained in memory during inference. 'Intr.' means that the method takes hours to plan and is not suitable for a real-world service. Bold text indicates the best result among methods. BFS cannot process inputs whose distance between a previous state and a next state is longer than two for the bulb. SMARTAID solves the action planning problem with the minimum number of visited states and states maintained in memory during training.

| Device | Distance | 1 | | 2 | | 3 | | 4 | | 5 | |
|---|---|---|---|---|---|---|---|---|---|---|---|
| | Method | Visited | Memory | Visited | Memory | Visited | Memory | Visited | Memory | Visited | Memory |
| Bulb | BFS | 186 | 183 | 17722 | 12227 | Intr. | Intr. | Intr. | Intr. | Intr. | Intr. |
| | A* | 369 | 369 | 586 | 584 | 742 | 739 | 856 | 852 | 926 | 921 |
| | Imp. A* | 477 | 372 | 766 | 606 | 946 | 765 | 1075 | 869 | 1145 | 927 |
| | SMARTAID | **78** | **74** | **116** | **90** | **138** | **100** | **156** | **105** | **162** | **106** |
| TV | BFS | 34 | 33 | 1162 | 336 | 4707 | 796 | 6326 | 966 | 9574 | 1213 |
| | A* | 59 | 60 | 140 | 140 | 169 | 169 | 189 | 189 | 209 | 209 |
| | Imp. A* | 147 | 63 | 253 | 156 | 298 | 189 | 365 | 241 | 361 | 237 |
| | SMARTAID | **26** | **23** | **55** | **47** | **61** | **50** | **63** | **51** | **66** | **52** |
| Robot cleaner | BFS | 8 | 5 | 50 | 17 | 126 | 29 | 170 | 37 | 236 | 40 |
| | A* | 16 | 17 | 25 | 25 | 35 | 34 | 42 | 39 | 49 | 45 |
| | Imp. A* | 26 | 18 | 51 | 32 | 88 | 50 | 116 | 64 | 160 | 82 |
| | SMARTAID | **6** | **3** | **6** | **3** | **7** | **4** | **7** | **4** | **7** | **4** |
| Air purifier | BFS | 2327 | 2052 | 2696 | 2294 | 2746 | 2492 | 2904 | 2413 | 2927 | 2445 |
| | A* | 6576 | 6575 | 6581 | 6580 | 6579 | 6578 | 6607 | 6606 | 6644 | 6643 |
| | Imp. A* | 8259 | 6005 | 8355 | 5949 | 8384 | 6041 | 8514 | 5839 | 8370 | 6103 |
| | SMARTAID | **1187** | **1184** | **1200** | **1196** | **1207** | **1203** | **1195** | **1191** | **1175** | **1172** |

In summary, SMARTAID efficiently finds an action sequence to set a device into the user's desired state. Note that SMARTAID does not require any additional information beyond the log data. This shows that SMARTAID offers an effective solution to plan action sequences even for unknown devices, instead of manually coding the action planner for each device which is the current solution in the IoT industry.

## Operation learning models

We compare various machine learning models for operation learning to find which model is the best option for SMARTAID. We divide capabilities into two groups: numerical and categorical ones. Numerical capabilities such as *'audioVolume'* or *'switchLevel'* have continuous values while categorical capabilities such as *'switch'* or *'cleaningMode'* have predefined classes as their values. Thus, numerical capabilities require regression methods while categorical capabilities require classification methods. We compare a linear regressor, a $k$-NN ($k$-nearest neighbors) regressor, a decision tree, and an MLP model for the regression methods. We compare a logistic regressor, a $k$-NN classifier, a decision tree, and an MLP model for the classification methods. Table 5 shows the accuracy of compared machine learning models for operation learning. Note that the overall accuracies of models are high since the real-world devices function according to their designed rules.

Only the linear model shows the maximum accuracy for all four devices among the regression methods. $k$-NN and decision tree show poor performance since the training data do not cover all possible values of a numeric capability and an argument when the range of possible values of the capability is wide. MLP performs worse than the linear model since its model capacity is too high compared to the complexity of the function of command. Meanwhile, all four classification methods achieve the best accuracies. This is because the training data cover all possible cases of categorical capabilities, unlike numerical capabilities.

**Table 5. Accuracy comparison between machine learning models to learn operations.**

| Device | Numerical capabilities | | Categorical capabilities | |
|---|---|---|---|---|
| | Method | Accuracy | Method | Accuracy |
| Bulb | Linear | 1.0000 | Logistic | 1.0000 |
| | k-NN | 0.5987 | k-NN | 1.0000 |
| | Decision tree | 1.0000 | Decision tree | 1.0000 |
| | MLP | 0.9591 | MLP | 1.0000 |
| TV | Linear | 0.9944 | Logistic | 1.0000 |
| | k-NN | 0.9792 | k-NN | 1.0000 |
| | Decision tree | 0.9792 | Decision tree | 1.0000 |
| | MLP | 0.9944 | MLP | 1.0000 |
| Robot cleaner | Linear | 0.7993 | Logistic | 0.9806 |
| | k-NN | 0.7551 | k-NN | 0.9879 |
| | Decision tree | 0.7708 | Decision tree | 0.9945 |
| | MLP | 0.7708 | MLP | 0.9890 |
| Air purifier | Linear | 1.0000 | Logistic | 1.0000 |
| | k-NN | 0.0200 | k-NN | 1.0000 |
| | Decision tree | 0.2008 | Decision tree | 1.0000 |
| | MLP | 0.5702 | MLP | 1.0000 |

Based on these results, SMARTAID chooses the best performing models (the linear regressor and the logistic regressor) for its operation learning models. This gives three major advantages to SMARTAID. First, these models are fast in terms of both training and inference. Second, their model sizes are smaller than other methods. Last, they are both interpretable models which give users a clear explanation of their decision. Thus, users easily understand the models, which helps usability in real-world applications.

In summary, SMARTAID effectively learns the operations of unknown devices even from raw log data without manual labeling. Thus, SMARTAID does not need further human efforts to design an appropriate action planner for each device based on its operation manual which is the common workflow in the IoT industry.

## Case study

We compare SMARTAID and other path-finding methods in special cases to show the strengths of SMARTAID. We experiment on two extreme cases: when there is no single command to change the value of any capability in the current state to that of the target state and when there is no setter command for a capability.

**Case 1: Indirect action sequences.** Assume that the current state of a TV is {*'switch':'off'*, *'volume':'20'*, . . .}and a user wants to change *'volume'* capability of the TV to *'0'* but keep the TV off after controlling it. Commands to control *'volume'* capability such as *'setVolume'*, *'volumeUp'*, and *'volumeDown'* are executable only if *'switch'* capability is set to *'on'*. Hence, there is no single command that changes *'volume'* of the TV closer to the target state. Thus, a smart home system needs to generate an indirect action sequence that turns on the switch to call the volume-controlling commands and then turns off the switch as shown in Fig 8. To compare the efficiency of SMARTAID and baselines for this case involving indirect action sequence, we run each method 1000 times where the value of *'switch'* capability is *'off'* both in the current state and the target state while values of *'volume'* are randomly selected in the two states.

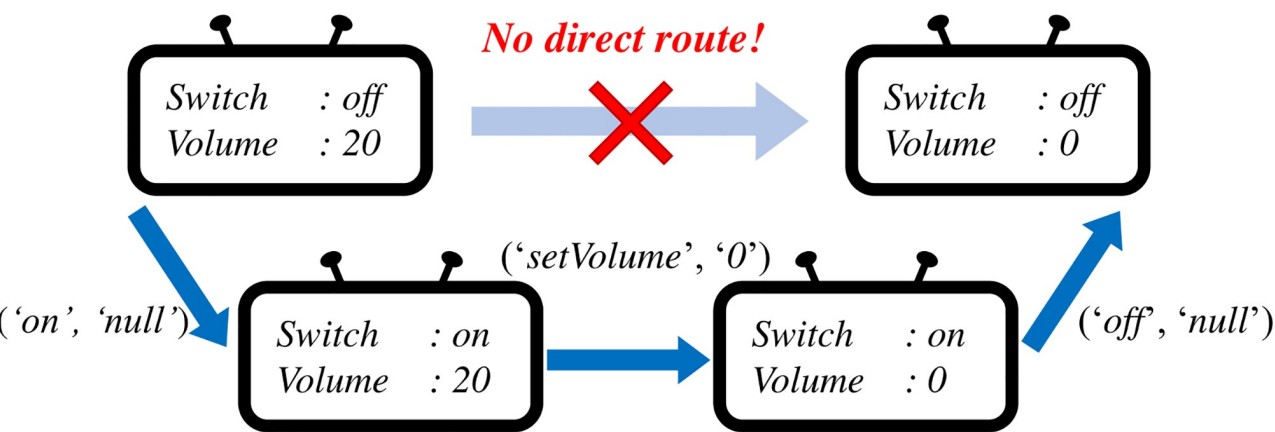

**Fig 8. Example of indirect action sequences.** A smart home system needs to find an indirect route to reach a target state from a current state.

Table 6 shows the results. SMARTAID shows the best efficiency among the methods in terms of both time complexity and space complexity. This is because SMARTAID finds the next target capability to plan the sequence from the learned dependency. Fig 9 shows the action planning process of each method. Baselines start with visiting adjacent states of the current state. Since all reachable states are farther from the target state, they further visit the 2-hop neighbors of the start state. In this process, they do not consider the efficiency of an action to reach the target state considering the capability it changes. Thus, the total number of visiting states exponentially grows consuming a lot of time and memory. However, SMARTAID first targets *'volume'* capability and generates an action sequence to set the target capability. Then, SMARTAID focuses on *'switch'* capability whose value should be *'on'* to meet the prerequisite condition of the already found action sequence. This procedure shows that SMARTAID targets capabilities one by oneso it does not visit unnecessary states and minimizes the cost of action planning.

**Case 2: No-setter action sequences.** Assume that there is no *'setVolume'* command for an old TV and the system needs to control *'volume'* capability with only *'volumeUp'* and *'volumeDown'* commands. If a user wants to change *'volume'* capability of the TV from *'10'* to *'50'*, the system needs to call *'volumeUp'* command 40 times as shown in Fig 10. We compare the efficiency of SMARTAID and baselines for this extreme case.

Table 7 shows the results. SMARTAID achieves the lowest time and memory usage among competitors. Fig 11 illustrates the searching process of the BFS, A*, improved A*, and SMARTAID. Other baselines try to visit every reachable state from the current state, which leads to exponentially growing costs as the distance increases. However, SMARTAID focuses only on *'volume'* capability and searches for the target state from the volume axis. Thus, SMARTAID generates an action sequence more efficiently than other methods.

**Table 6. Efficiency comparison of indirect action sequence planning.** SMARTAID visits and memorizes lesser states during inference compared to other methods.

| Method | BFS | A* | Imp. A* | SMARTAID |
|---|---|---|---|---|
| # Visited states | 6814 | 261 | 437 | 56 |
| # Maintained states | 1548 | 260 | 278 | 51 |

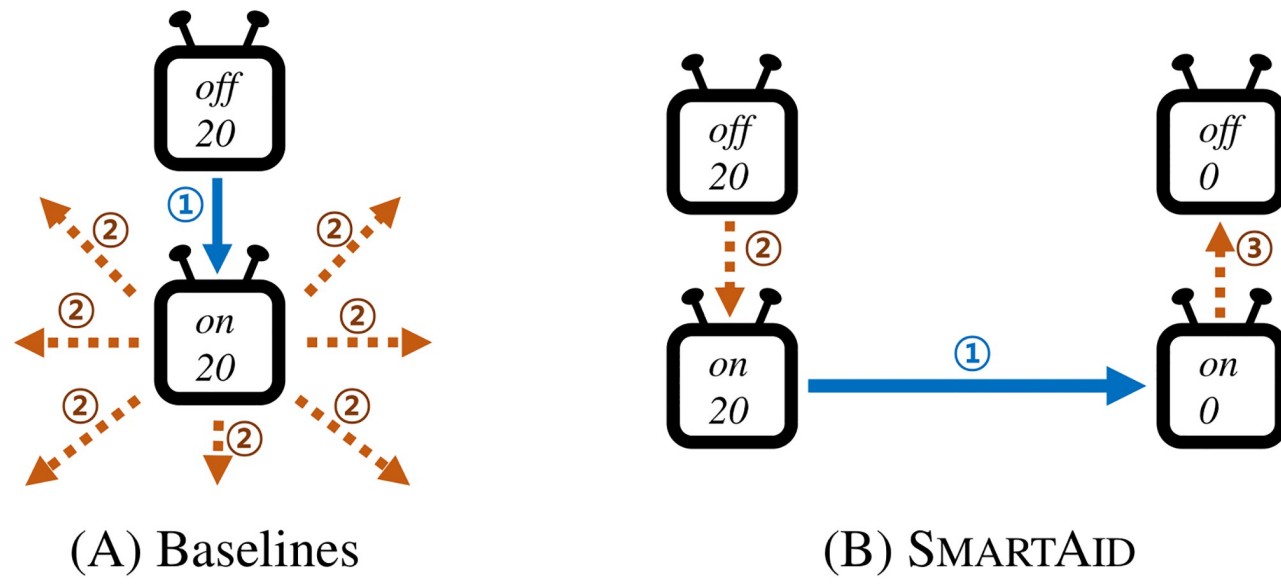

**Fig 9.** Illustrations of the searching process of BFS, A\*, and improved A\* (left), and SMARTAID (right). Numbers on arrows indicate the search order of each method. Baselines find the target state from the neighbors of visited states, while SMARTAID first constructs a sequence to set the target capability and refines the action sequence to be valid by Algorithm 2. Hence, SMARTAID plans the actions more efficiently with lesser amounts of visits compared to other methods.

## Related work

### Action recommendation for smart home

Recently there have been studies about predicting the next action of IoT devices desired by a user as the IoT industry develops. Action recommendation for smart home aims to recommend actions to users when controlling their devices at home. [3, 4] try to capture the user intention from the previous history of the user and select the most appropriate action for the intention. SmartSense utilizes self-attention layers and common sense knowledge to extract users' intentions from a given session [3]. DeepUDI further employs multi-modal embedding and an intent-aware encoder to interpret users' intention behind the given action history [4]. However, these studies focus on predicting the following single action that represents the user's unknown intention after the session rather than finding an action sequence to fulfill the user's need. Action planning problem for smart home focuses on finding a sequence of actions when a user's intention is given.

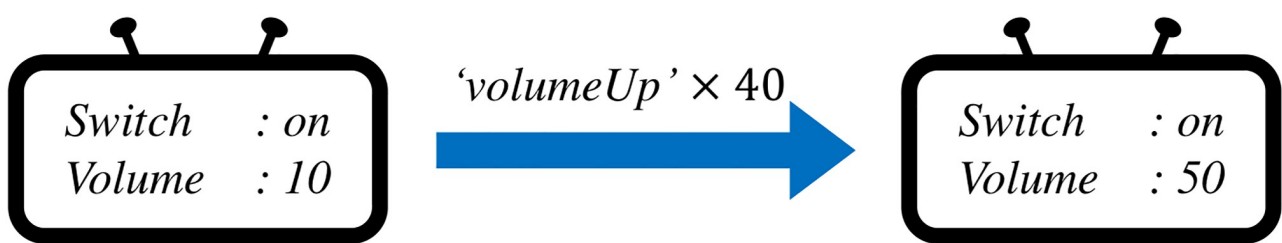

**Fig 10. Example of no-setter action sequences.** Even the shortest action sequence needs 40 actions to fulfill the user's query since there is no direct setter command for *'volume'* capability.

**Table 7. Efficiency comparison for no-setter action sequences.** SMARTAID visits and memorizes lesser states during inference compared to other methods.

| Method | BFS | A* | Imp. A* | SMARTAID |
|---|---|---|---|---|
| # Visited states | 132089 | 11681 | 2282 | **103** |
| # Maintained states | 6559 | 837 | 1978 | **51** |

## Path planning

Finding a path to the target point is a widely researched topic in the robotics area [11, 12]. The path planning problem aims to find a path on the given map from the starting point to the target point. It is crucial to find an optimal path between two points in the search space to control a robot [13, 14] or design an autonomous vehicle [15]. A lot of approaches such as neural networks [16, 17], fuzzy logic [18], genetic algorithms [19], and various heuristics [12, 20, 21] are studied to solve the path planning problem. Moreover, recent studies in this area aim to consider physical factors such as battery [22], rotation [23], or other vehicles [24]. However, these studies mainly focus on avoiding obstacles in the geometric search space with complete descriptions of actions, and do not consider the relationships between the capabilities and commands of IoT environments.

There are also studies of path planning related to IoT. For instance, EMESP [25] utilizes the Bug algorithm to find an efficient path for data transmission in a wireless environment. Follow Me-AI [26] manages resources of the smart environment for a user. However, these studies focus on effectively handling the given smart environment and do not consider scenarios involving unknown devices. Action planning problem for smart home needs to capture the information of a new device from a partial state transition graph and consider the nature of IoT environments rather than the geometric obstacles or physical costs.

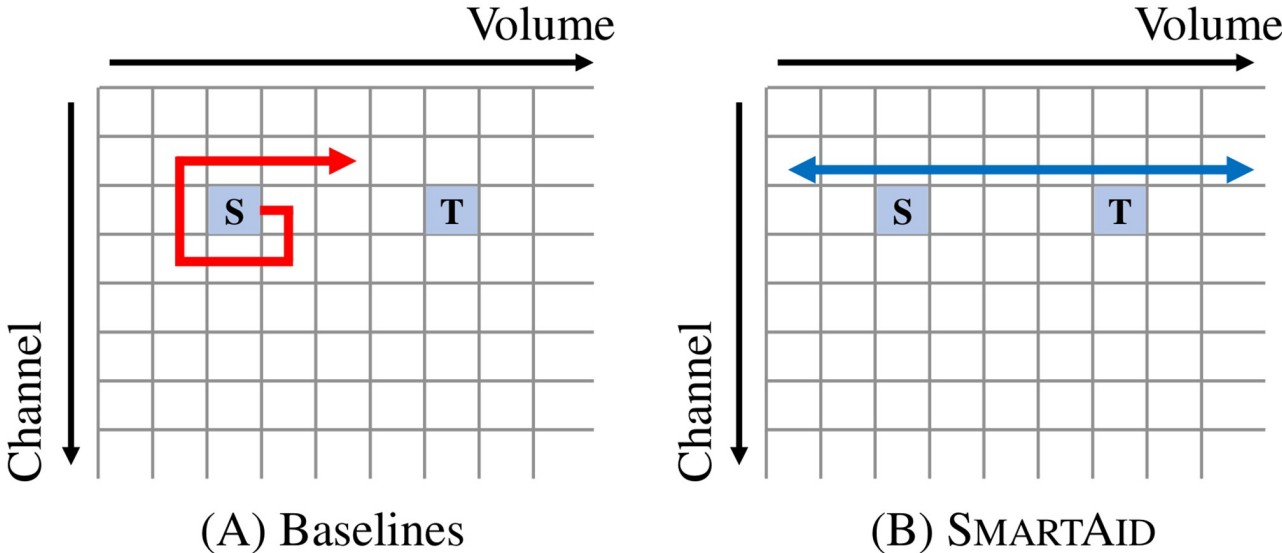

**Fig 11.** Illustrations of search spaces of baselines (left) and SMARTAID **(right)**. 'S' and 'T' indicate the start state and the target state. Other baselines such as BFS need to find the target state from the entire grid which consumes massive time and memory. Meanwhile, SMARTAID limits the search space to the volume axis since SMARTAID focuses on a single capability at once. Thus, SMARTAID reaches the target state faster with lower memory usage.

### Shortest path

Finding the shortest path is a deeply researched topic in graph theory. The goal is to find the shortest or near-shortest path between two points on a given graph. There are two kinds of shortest path methods: all pairs shortest path (APSP) and N pairs shortest path (NPSP). APSP algorithms aim to find paths between all possible pairs of nodes on the given graph [27, 28]. NPSP algorithms focus on finding paths between N given pairs of nodes on the graph [29]. Action planning problem for smart home is related to NPSP since it tries to find an action path between a source state and a target state on a state transition graph. However, since NPSP algorithms aim to achieve efficiency for a large N, they do not show a practical advantage over general search algorithms like BFS or $A^*$ in terms of complexity for a small N [29]. Thus, NPSP methods are inappropriate for the action planning problem for smart home since the scale of the state transition graph is massive compared to the number of queries as shown in Fig 2.

## Conclusion

In this paper, we propose SMARTAID, an efficient and accurate action planning method for smart home. SMARTAID manages an intractable state transition graph through a compact representation focusing on commands. SMARTAID collects the dependency and learns the function of each command from the state transition log of the device. Then, SMARTAID finds a sequence of actions to change the current state into the target state by sequentially setting up each capability with a condition-matching process. Extensive experiments show that SMARTAID is accurate, and the most efficient method to solve the action planning problem, reducing up to 85.7% visited states and up to 91.1% states maintained in memory during inference. SMARTAID also demonstrates the greatest scalability in various special cases of real-world action planning problems. This research introduces a solution that allows IoT systems to efficiently manage and control an ever-expanding array of smart devices, including previously unrecognized ones. Future works include further optimizing the action planning algorithm to find the shortest action sequence to process the user's query.

## Author Contributions

**Conceptualization:** Jongjin Kim, Jaeri Lee, Jeongin Yun, U. Kang.

**Data curation:** Jongjin Kim, Jaeri Lee, Jeongin Yun.

**Methodology:** Jongjin Kim.

**Project administration:** U. Kang.

**Software:** Jongjin Kim, Jaeri Lee, Jeongin Yun.

**Supervision:** U. Kang.

**Validation:** Jaeri Lee, Jeongin Yun, U. Kang.

**Writing – original draft:** Jongjin Kim.

**Writing – review & editing:** Jongjin Kim, Jaeri Lee, Jeongin Yun, U. Kang.

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
