## [Decision Letter · Decision Letter 0]

4 Mar 2024

PONE-D-24-02724Dependency-Aware Action Planning for Smart HomePLOS ONE

Dear Dr. Kang,

Thank you for submitting your manuscript to PLOS ONE. After careful consideration, we feel that it has merit but does not fully meet PLOS ONE’s publication criteria as it currently stands. Therefore, we invite you to submit a revised version of the manuscript that addresses the points raised during the review process.

We look forward to receiving your revised manuscript.

Kind regards,

Praveen Kumar Donta, Ph.D.

Academic Editor

PLOS ONE

Reviewers' comments:

Reviewer's Responses to Questions

**Comments to the Author**

1. Is the manuscript technically sound, and do the data support the conclusions?

Reviewer #1: Partly

Reviewer #2: Yes

Reviewer #3: Yes

2. Has the statistical analysis been performed appropriately and rigorously? 

Reviewer #1: N/A

Reviewer #2: Yes

Reviewer #3: Yes

3. Have the authors made all data underlying the findings in their manuscript fully available?

Reviewer #1: Yes

Reviewer #2: Yes

Reviewer #3: Yes

4. Is the manuscript presented in an intelligible fashion and written in standard English?

Reviewer #1: Yes

Reviewer #2: Yes

Reviewer #3: Yes

5. Review Comments to the Author

Reviewer #1: In this paper, the authors present a Machine Learning-based action planning method for IoT smart homes that reduces the state transition graph and generates an effective sequence of commands to reach the desired state without side effects. The paper is easy to read, and the proposed method is available on GitHub, which increases the value of the work. However, the paper has some drawbacks, and I am not convinced enough that this research problem is worth the effort. My comments that should be addressed are the following:

The paper does not discuss the effect of the results. Indeed, the proposed method visits fewer states, consumes less memory, and is, all in all, a time-efficient and memory-efficient method. But what exactly do the results mean in terms of performance for a real-world service? What about the execution speed of the given command? Does a smart light bulb really need this effort?

The different ML models reached 100% accuracy, which is rare in the field of ML, and it can suggest an oversimplified model or overfitting. How can the readers be sure that the model is general enough?

The paper deals with the SmartThings API, commands and logs, however, a general introduction on how the platform works is missing. What about the compatibility and usability of the proposed method? What kind of requirements are needed to utilise the method in the field of IoT healthcare, for example? How were the log files gathered and why are they not readable on GitHub?

To measure the contribution of this work, it is also important to understand the current operation of such devices. It is stated in the text that the state transition graphs of real-world devices are intractable and cannot be stored. Then, how exactly do these devices work nowadays?

Sometimes I felt the given examples were odd. For instance, when the current state of a TV is off and the user wants to change volume but keeps the TV off after controlling it. It does not seem like a realistic scenario for me. On the other hand, it is also stated that the proposed method merges capabilities that form a cycle into a single capability to guarantee that there exists a topological order of capabilities. That is an interesting phenomenon, which should be presented in more detail and as a scenario as well.

The related work section is too brief, less than a page.

The figures should be inserted in the text instead to the end of the paper.

Possible future directions are missing from the conclusion of the paper.

The term ‘prerequisite conditions’ sounds strange to me. Prerequisite is a synonym of precondition or prior condition.

Reviewer #2: The paper contains interesting work and publishable material. However, the following points should be incorporated in the paper for improvement:

1. The contribution of the proposed work is lacking in the manuscript. The main contribution should be highlighted at the end of the introduction section in bullet form for easiness of the readers.

2. The literature review of existing work is limited. Moreover, there should be a paragraph in the introduction section that should highlight the limitations of existing techniques that lead to the motivation of the proposed work.

3. More explanation and analysis of the achieved results should be added in the numerical results section. Moreover, a comparison of the proposed work with existing techniques should be added.

Reviewer #3: - The use cases of the research proposal are well explained with examples to understand.

- The research propose SmartAid, an efficient and accurate method to solve the action planning problem for home devices.

- Algorithms for the proposal as explained with the examples.

- The research results are provided with datasets and its results.

Review Comments :

- What is the advancement of this research activity in near future ? Please mention in the conclusion section.

- What are commercial devices used in the research and its results ?

- What are different types of smart homes devices that be applied and used in the research ? Does the results same irrespective of devices ?

- What are current literature research and its results ? How this research results are compared with the literature or commercial similar solutions ?

6. PLOS authors have the option to publish the peer review history of their article (what does this mean?). If published, this will include your full peer review and any attached files.

Reviewer #1: No

Reviewer #2: **Yes: **Muhammad Imran

Reviewer #3: **Yes: **Deepak Dasaratha Rao

---

## [Author Response · Author response to Decision Letter 0]

15 Apr 2024

We would like to thank the reviewers for their high quality reviews and constructive comments. Below, we summarize how we addressed the reviewers’ requirements.

1. Reviewer 1.

• (R1-1) The paper does not discuss the effect of the results. Indeed, the proposed method visits fewer states, consumes less memory, and is, all in all, a time-efficient and memory-efficient method. But what exactly do the results mean in terms of performance for a real-world service? What about the execution speed of the given command? Does a smart light bulb really need this effort?

– (A1-1) We add the detailed comparison with SmartAid and current IoT industry in the experiment section from lines 474 to 478 and lines 506 to 509. Currently, developers need to manually design an action planner for each device to make the system able to control the device. On the other hand, our research offers the solution to automatically learn the operations of unknown devices and plan appropriate action sequences to meet the user’s needs. Furthermore, an efficient action planning process lets SmartAid handle more advanced devices with numerous capabilities and complex dependencies.

• (R1-2) The different ML models reached 100% accuracy, which is rare in the field of ML, and it can suggest an oversimplified model or overfitting. How can the readers be sure that the model is general enough?

– (A1-2) This experiment focuses on learning the operation of real-world devices, which operate according to the predefined rules made by manufacturers. Hence, if the rules are simple enough to be fully represented by machine learning models, models achieve 100% accuracy as long as there are no malfunctions in the device while collecting logs. For instance, a linear regression model would accurately learn the function of ‘volumeUp’ command, represented as y = x + 1. However, it would be challenging for a machine learning model to fully learn the function of a command if the logs are unclear due to the physical limitations or the complex dependencies between capabilities. This is the case with the robot cleaner in our experiment.

• (R1-3) The paper deals with the SmartThings API, commands and logs, however, a general introduction on how the platform works is missing. What about the compatibility and usability of the proposed method? What kind of requirements are needed to utilise the method in the field of IoT healthcare, for example? How were the log files gathered and why are they not readable on GitHub?

– (A1-3) There are various smart home systems such as SmartThings, Google Alexa, and Apple Home. However, despite terminological differences, the framework of representing the state of a device through various capabilities and their values, and sending actions to control the devices remains the same across these platforms. We add this explanation about the general framework of smart home systems in lines 113 to 116. Since our algorithm does not need further information about a device other than its log data, the proposed method would be applicable to other fields that operate within a framework where a system has access to read a device’s state and send desired actions to control the device. We collect log data by repeatedly sending random commands and checking the change in the device’s state as mentioned in line 425. We uploaded the datasets in the form of Pickle binary files so they may not be readable on the browser.

• (R1-4) To measure the contribution of this work, it is also important to understand the current operation of such devices. It is stated in the text that the state transition graphs of real-world devices are intractable and cannot be stored. Then, how exactly do these devices work nowadays?

– (A1-4) Currently, device manufacturers provide solutions for integration with systems like SmartThings, or smart home system providers offer solutions for their products or those of their partners. Either way, developers need to manually code how to handle those devices directly. However, our research offers a solution to handle a newly connected device without developers intervening for integration since the hub itself would learn the operation of the new device.

• (R1-5) Sometimes I felt the given examples were odd. For instance, when the current state of a TV is off and the user wants to change volume but keeps the TV off after controlling it. It does not seem like a realistic scenario for me. On the other hand, it is also stated that the proposed method merges capabilities that form a cycle into a single capability to guarantee that there exists a topological order of capabilities. That is an interesting phenomenon, which should be presented in more detail and as a scenario as well.

– (A1-5) The example may seem a bit unusual since we try to explain all examples in the paper with the TV for simplicity. However, the scenario of turning on a device, operating it, and then turning it off is a reasonable scenario for other devices. For instance, one could want to activate the robot, move it to the other room, and then return it to standby mode. We add more explanation about merging capabilities from lines 310 to 315.

• (R1-6) The related work section is too brief, less than a page.

– (A1-6) To the best of our knowledge, this is the first study addressing the action planning problem for smart home. Hence, we review studies of related fields that cover more general pathfinding problems. However, those studies are not directly applicable to our problem as mentioned at the end of each subsection of related works.

• (R1-7) Possible future directions are missing from the conclusion of the paper.

– (A1-7) We add the possible future works at the end of the conclusion from lines 604 to 608.

• (R1-8) The term ‘prerequisite conditions’ sounds strange to me. Prerequisite is a synonym of

precondition or prior condition.

– (A1-8) We choose the term ‘prerequisite’ to emphasize that the prerequisite condition is a requirement to execute a specific action, rather than just an observed previous state before execution.

2. Reviewer 2.

• (R2-1) The contribution of the proposed work is lacking in the manuscript. The main contribution should be highlighted at the end of the introduction section in a bullet form for easiness of the readers.

– (A2-1) We summarized the contributions of our paper in bullet form from lines 59 to 81.

• (R2-2) The literature review of existing work is limited. Moreover, there should be a paragraph in the introduction section that should highlight the limitations of existing techniques that lead to the motivation of the proposed work.

– (A2-2) To the best of our knowledge, this is the first study addressing the action planning problem for smart home. Hence, we introduce related fields that cover more general pathfinding problems. However, those studies are not directly applicable to our problem as mentioned in the related works section. Currently, developers need to manually code the solution to handle the action planning problem for each device. However, this is getting impractical since the IoT industry is growing rapidly and the number of smart devices is increasing explosively as we mentioned in the introduction section. Thus, our study aims to provide an automated solution to plan action sequences even for unknown devices. We emphasize this motivation in the introduction from lines 12 to 14.

• (R2-3) More explanation and analysis of the achieved results should be added in the numerical results section. Moreover, a comparison of the proposed work with existing techniques should be added.

– (A2-3) We add the detailed comparison with SmartAid and current IoT industry in the experiment section from lines 474 to 478 and lines 506 to 509. Currently, developers need to manually design an action planner for each device to make the system able to control the device. On the other hand, our research offers the solution to automatically learn the operations of unknown devices and plan appropriate action sequences to meet the user’s needs.

3. Reviewer 3.

• (R3-1) What is the advancement of this research activity in near future? Please mention in the conclusion section.

– (A3-1) We add the possible future works at the end of the conclusion from lines 604 to 608.

• (R3-2) What are commercial devices used in the research and its results?

– (A3-2) We add the exact names of devices used in the research to the experimental section from lines 422 to 424.

• (R3-3) What are different types of smart homes devices that be applied and used in the research? Does the results same irrespective of devices?

– (A3-3) There are various smart home systems such as SmartThings, Google Alexa, and Apple Home. However, despite terminological differences, the framework of representing the state of a device through various capabilities and their values, and sending actions to control the devices remains the same across these platforms. We add an explanation of the general framework of smart home systems in lines 113 to 116. Considering that our algorithm operates solely on logs without requiring additional information about the devices, the proposed method would accurately perform regardless of the system or type of device.

• (R3-4) What are current literature research and its results? How this research results are compared with the literature or commercial similar solutions?

– (A3-4) To the best of our knowledge, this is the first study to address the action planning problem for smart home. We review the related research topics that cover the general pathfinding problems but they are not applicable to this problem as mentioned in the related work section. The current commercial solution is to manually code the program to plan action sequences for each device. However, this is becoming impractical due to the increasing number of smart devices and the growing industry. Our study aims to provide an automated solution for the action planning problem even for unknown devices.

---

## [Decision Letter · Decision Letter 1]

13 May 2024

PONE-D-24-02724R1Dependency-Aware Action Planning for Smart HomePLOS ONE

Dear Dr. Kang,

Thank you for submitting your manuscript to PLOS ONE. After careful consideration, we feel that it has merit but does not fully meet PLOS ONE’s publication criteria as it currently stands. Therefore, we invite you to submit a revised version of the manuscript that addresses the points raised during the review process.

We look forward to receiving your revised manuscript.

Kind regards,

Praveen Kumar Donta, Ph.D.

Academic Editor

PLOS ONE

Journal Requirements:

Additional Editor Comments:

The authors are requested to address the following comments before it is considered for publication.

1. There are some language corrections in the paper, please proofread carefully.

2. Provide the source codes related to the papers through GITHUB links.

3. There are several path planning algorithms which are published recently for IoT/WSNs while considering obstacles.

The authors must provide them more carefully.

For example, Bug2 algorithm-based data fusion using mobile element for IoT-enabled wireless sensor networks,

and smart environments related such as Follow-Me AI: Energy-Efficient User Interaction with Smart Environments

4. Provide computational complexity for the algorithms 1&2 must be evaluated.

Reviewers' comments:

Reviewer's Responses to Questions

**Comments to the Author**

1. If the authors have adequately addressed your comments raised in a previous round of review and you feel that this manuscript is now acceptable for publication, you may indicate that here to bypass the “Comments to the Author” section, enter your conflict of interest statement in the “Confidential to Editor” section, and submit your "Accept" recommendation.

Reviewer #1: All comments have been addressed

Reviewer #3: All comments have been addressed

2. Is the manuscript technically sound, and do the data support the conclusions?

Reviewer #1: Yes

Reviewer #3: Yes

3. Has the statistical analysis been performed appropriately and rigorously? 

Reviewer #1: Yes

Reviewer #3: Yes

4. Have the authors made all data underlying the findings in their manuscript fully available?

Reviewer #1: Yes

Reviewer #3: Yes

5. Is the manuscript presented in an intelligible fashion and written in standard English?

Reviewer #1: Yes

Reviewer #3: Yes

6. Review Comments to the Author

**Reviewer #1:** (No Response)

**Reviewer #3:** All the review comments updated in the revised paper. Thank you for updating and sharing for the review.

7. PLOS authors have the option to publish the peer review history of their article (what does this mean?). If published, this will include your full peer review and any attached files.

Reviewer #1: No

Reviewer #3: **Yes: **Deepak Dasaratha Rao

---

## [Author Response · Author response to Decision Letter 1]

28 May 2024

We would like to thank the editor for your high quality reviews and constructive comments. Below, we summarize how we addressed your requirements.

• (R1) There are some language corrections in the paper, please proofread carefully.

- (A1) We double-checked the entire manuscript for language errors and made the necessary corrections.

• (R2) Provide the source codes related to the papers through GITHUB links.

- (A2) We denote the link of our codes and datasets at the end of introduction section in lines 70

and 81.

• (R3) There are several path planning algorithms which are published recently for IoT/WSNs while considering obstacles. The authors must provide them more carefully. For example, Bug2 algorithm-based data fusion using mobile element for IoT-enabled wireless sensor networks, and smart environments related such as Follow-Me AI: Energy-Efficient User Interaction with Smart Environments

- (A3) We add mentioned papers and summarize diffrence of our work compared to previous works of path planning from lines 580 to 583. Our paper focuses on learning and training an unknown device with its state transition log data, while previous works of path planning aim to effectively control already given smart environments.

• (R4) Provide computational complexity for the algorithms 1&2 must be evaluated.

- (A4) We analyze the complexity of algorithms in Complexity Analysis section (line 374). Theorems 1 and 2 analyze the time complexity and the space complexity of Algorithm 1, respectively. Note that we did not separately analyze the complexity of Algorithm 2 which is a subroutine of Algorithm 1.

---

## [Editor Report · Decision Letter 2]

30 May 2024

Dependency-Aware Action Planning for Smart Home

PONE-D-24-02724R2

Dear Dr. Kang,

We’re pleased to inform you that your manuscript has been judged scientifically suitable for publication and will be formally accepted for publication once it meets all outstanding technical requirements.

Kind regards,

Praveen Kumar Donta, Ph.D.

Academic Editor

PLOS ONE